# Cross Encoder-Decoder Transformer with Global-Local Visual Extractor for Medical Image Captioning

**DOI:** 10.3390/s22041429

**Published:** 2022-02-13

**Authors:** Hojun Lee, Hyunjun Cho, Jieun Park, Jinyeong Chae, Jihie Kim

**Affiliations:** 1Department of Computer Science and Engineering, Dongguk University, Seoul 04620, Korea; cajun7@dgu.ac.kr (H.L.); chohyunjun1111@gmail.com (H.C.); 5656jieun@dgu.ac.kr (J.P.); 2Department of Artificial Intelligence, Dongguk University, Seoul 04620, Korea; jiny419@dgu.ac.kr; 3Okestro Ltd., Seoul 07326, Korea

**Keywords:** medical image captioning, deep learning, transformer

## Abstract

Transformer-based approaches have shown good results in image captioning tasks. However, current approaches have a limitation in generating text from global features of an entire image. Therefore, we propose novel methods for generating better image captioning as follows: (1) The Global-Local Visual Extractor (GLVE) to capture both global features and local features. (2) The Cross Encoder-Decoder Transformer (CEDT) for injecting multiple-level encoder features into the decoding process. GLVE extracts not only global visual features that can be obtained from an entire image, such as size of organ or bone structure, but also local visual features that can be generated from a local region, such as lesion area. Given an image, CEDT can create a detailed description of the overall features by injecting both low-level and high-level encoder outputs into the decoder. Each method contributes to performance improvement and generates a description such as organ size and bone structure. The proposed model was evaluated on the IU X-ray dataset and achieved better performance than the transformer-based baseline results, by 5.6% in BLEU score, by 0.56% in METEOR, and by 1.98% in ROUGE-L.

## 1. Introduction

Image captioning is a task that automatically generates a description of a given image. In the medical field, the technique can be used to generate medical reports from X-ray or CT images. A model that automatically generates reports on medical images can assist doctors to focus on notable image areas or explain their findings. It can also help reduce medical errors and reduce costs per test.

The medical image reports include an overall description of the image as well as an explanation of the features of the suspected disease. In addition, similar patterns such as judgement of pleural effusion, pneumothorax, lung, heart, and skeletal structural abnormalities exist in the reports. There has been a lot of research in automatic generation of medical image descriptions. For example, the relational memory and memory-driven conditional layer network on the standard transformer (R2Gen) were proposed to generate reports of chest X-ray images [1]. However, sufficient descriptions about global features such as bone structure, or more detailed size information that spreads over a global image, have not existed in previous studies. This is because the inputs of the transformer-based model are generated by splitting the image into patch pieces. Through this process, global information is lost and weaknesses are shown in generating features requiring more than one patch. Figure 1 shows an example of such a shortage. The method created the descriptions of the features well, but omitted some information about the size of an organ or bone structure that needs to be judged from the overall understanding of the image. As examples of this issue, there are the terms “the heart is not significantly” and the difference between “arthritic changes” and the inaccurate “no acute” expressions, as shown in Figure 1. Because these shortcomings are important in medical image captioning, we aimed to address them.

In this paper, we propose the Global-Local Visual Extractor (GLVE) and the Cross Encoder-Decoder Transformer (CEDT). The GLVE captures local features as well as global features in images. By extracting both global and local features with the GLVE, our model learns the organ size, skeletal structure, and irregular lesion area.

In addition, the CEDT uses all levels of information of the encoders in order not to lose features extracted from the GLVE when decoding phrases. In other words, the CEDT additionally hands over low-level encoder information representing the images to avoid losing the features extracted by the GLVE into the decoder. By using the CEDT, the explanation of the parts that should be judged by the overall understanding of the image can be generated effectively. We evaluated the performance of the methods on the IU X-ray dataset. The contributions of this paper can be summarized as follows:We introduce a GLVE to extract both global and local visual features. To get a local region, we construct a mask to crop the global image with the attention heat map generated from the global visual extractor. By using the GLVE, our model learns from both the global image and the local region that is the disease-specific region.We propose a CEDT to generate captions by adding the encoder’s low-level features and high-level features without missing the encoder’s features. The CEDT uses both low-level features representing images present in the low-level encoders and abstracted information present at high levels to show good results for not only local features but also global features.Evaluating the proposed method on the IU X-ray dataset, we achieved better performance than the baseline results, by 5.62% in BLEU score, by 0.56% in METEOR, and by 1.98% in ROUGE-L.

## 2. Related Work

Research on image captioning, which is most relevant to the present work, is actively underway [2,3,4]. An encoder-based multimodal transformer model [5] represents inter-sentence relationships and introduces multi-view learning to provide a variety of visual representations. Recently, a variety of new models have been proposed to obtain global information. Among them, there are ways to use the information of each layer of the encoder [6,7]. A meshed transformer with memory [6] was used to learn and better understand the context in the image that is difficult to judge simply by part. In addition, decoders took all outputs of meshed memory encoders, which inspired an idea to improve our model. A global enhanced transformer [7] showed performance improvements through learning global representations using feature information of each encoder.

In addition, studies on medical image captioning also have been active [8,9,10]. Baoyu et al. [11] introduced a co-attention (CoATT) mechanism for localizing sub-regions in the image and generating the corresponding descriptions, as well as a hierarchical LSTM model to generate long paragraphs. In addition, the cooperative multi-agent system (CMAS-RL) [12] was proposed for modeling the sentence generation process of each section. Chen et al. [1] proposed a memory-driven transformer structure. Relational memory was used to remember the previous generation process and was integrated into the transformer using the memory-driven conditional layer network. Guan et al. [13] proposed an attention-guided two-branch convolution neural network (AG-CNN) model to improve noise reduction and alignment. After learning about a specific area for analyzing each disease through a local branch, the information lost by the local branch was compensated by integrating the global branch into it. By incorporating this into the GLVE module of our model, both global and local features can be extracted from images. In addition, inspired by Chen et al. [1], we added the relational memory and memory-driven conditional layer to our model for reflecting captioning of similar images.

## 3. Method

### 3.1. Global-Local Visual Extractor

We encoded a radiology image into a sequence of patch features with the convolution neural network (CNN)-based feature extractor. Given a radiology image like top row of Figure 2, the encoded patch features are the output of the global visual extractor. Global visual features are defined as follows:(1)Xglobal=x1global, x2global, …, xsglobal,
where xiglobal∈ℝd is the patch feature of the image, for i∈1, 2, …, s, s denotes the numbers of patches in the image and d is the size of the feature vector, which are extracted from the global visual extractor of the GLVE.

There are problems that occur when learning only from these global images: (1) Since the size of the lesion area may be small or the location may not be predicted, the global visual features may include the information on noise. (2) There are various capture conditions such as the patient’s posture and body size [13]. To address the problems, the local visual extractor is added with the global visual extractor, and the features extracted from the local visual extractor are concatenated to the global visual features. As shown in Figure 3, we crop the important part of the image with the last layer of the global visual extractor, resize it to the same size as the image and use it as the input of the local visual extractor. We create a binary mask to locate the region by applying thresholds on the feature maps. In the kth channel of the last CNN layer of the global visual extractor, $f_{k}\left(x,y\right)$ denotes the activation values of spatial location $\left(x,y\right)$, where $k\in \left\{1,\;\ldots ,\;K\right\}$, *K* = 2048 in Resnet-101, and $\left(x,y\right)$ represents the coordination of the feature map. The attention heat map is defined as:(2)$$H_{g}\left(x,y\right)=\underset{k}{\max}\left(\left|f_{g}^{k}\left(x,y\right)\right|\right),$$
indicating the importance of activations. A binary mask, M, designed for locating the regions with activation values larger than threshold τ, is defined as:(3)$$M\left(x,y\right)=\left\{\begin{array}{c} 1,\;\;\;\;H_{g}\left(x,y\right)>\tau \\ 0,\;\;\;\;\mathrm{otherwise}\; \end{array}\right..$$

We find a maximum region that covers the discriminative points with the mask M to crop the most important part of a given image that needs to be focused on. The region is cropped from an original image, and we use it as the input of the local visual extractor as shown in Figure 3. Local visual features are defined as:(4)Xlocal=x1local, x2local,…, xslocal,
where xslocal∈ℝd is the patch feature of the cropped image, which are extracted from the local visual extractor. The global visual extractor can then extract the global features so that they accurately capture the size of an organ or bone structure. Additionally, the local visual extractor reduces noise and provides improved alignment. Xglobal_local, the concatenation of Xglobal and Xlocal, are used as the input of the encoder of the CEDT as shown in Figure 3. For better extraction of both global and local features of the images, we pre-train the GLVE with Chest X-ray14.

### 3.2. Cross Encoder–Decoder Transformer

The transformer-based image captioning model has shown good performance in long text descriptions by capturing the association between image features more than other encoder–decoder models [1,5,6,7]. This is because the transformer is an architecture that adopts the attention mechanism. Attention is a mechanism of focusing on important words by determining related parts to correct poor performance in situations where input sentences are long. There are several attention functions, such as additive attention and dot-production attention. Among them, scaled dot-product attention is the most commonly used because it is much faster, more space-efficient, and can prevent exploding into huge values by scaling. Scaled dot-product attention [14] is defined as:(5)$$Attention\left(Q,K,V\right)=softmax\left(\frac{QK^{T}}{\sqrt{d_{k}}}\right),$$
where Q denotes a matrix where queries are packed into, K denotes a matrix where keys are packed into, V denotes a matrix where values are packed into, and $d_{k}$ is the dimension of queries and keys. The attention function finds the similarity among all keys for a given query, respectively. Then, this similarity is reflected in each value mapped to the key using the similarity as the weight, while the attention value is the weighted sum of all values. Transformer based on attention mechanism showed good performance for local features such as one organ in medical image captioning. However, there has been a lack of text generation for global features that cannot be known only from information of one part in the image, and many studies have been aimed at improving the performance of global features. We also found a problem of lack of text generation for global information, such as bone structure and the detailed difference in the size of organs (e.g., enlargement), in the field of image captioning and performing better image captioning by inserting global information.

We propose the CEDT to obtain both low and high-level information in an image based on R2Gen [1]. The CEDT connects between layers of the transformer, as shown in Figure 4. The standard transformer uses only the last layer information of the encoder, but we also use low features additionally to avoid losing information on the input image features of the encoders. We modify it by using the idea of meshed-memory paper [6], which stores all the encoder’s outputs and uses it in the decoder. We use the outputs of all the encoders on each decoder with multi-head attention and weighted-sum, as shown in the gray-dashed box in Figure 4. For these values, parallel multi-head attention is performed within each decoder to obtain all-level information of the encoder. In addition, each result of multi-head attention is a weighted sum with parameters α, β, and γ. These parameters are used to control the amount of information in low-level features and high-level features extracted from the encoder. The connection between the encoder and decoder layers makes use of more information from the image features in the decoding process. This is because different levels of abstraction information for image features existing in each encoder layer are used in this structure. As a result, the model generates better captioning by adding the features extracted from each layer of the encoder than the predicted caption of the baseline.

### 3.3. Image Captioning with GLVE and CEDT

We propose a new image captioning model with GLVE and CEDT, as shown in Figure 5. GLVE and CEDT are combined as GLVE features are used as the input of the first encoder of CEDT. We also make use of relational memory (RM) and memory-driven conditional layer normalization (MCLN) of Chen et al. [1] for recording and utilizing the important information. Through this model, we aim to obtain both local feature and global feature information with the GLVE and various abstraction information of images with the CEDT, so that text generation for the global feature is also improved. We use RM and MCLN because they are suitable and effective methods for medical image captioning. RM learns and remembers the important key feature information and patterns of radiology reports, and MCLN connects memory to the transformer on the decoding process. Through these modules, the reports are generated containing sufficient important features. As we propose the GLVE, the visual extractor can extract both global and local features to capture the size of an organ or bone structure and disease-specific features. The CEDT uses various levels of abstract information from the encoder to avoid losing GLVE output features. Through these modules, our model shows significant performance improvement.

## 4. Experiments

### 4.1. Implementation Details

Our model was trained on a single NVIDIA RTX 3900 with a batch size of 16 using cross entropy loss and the Adam optimizer with gamma of 0.1. The learning rate of the GLVE was set to 5 × 10^−4^ and step size 50, and the learning rate was adjusted every 50th step. We kept the size of the transformer, and all details of RM and MCLN unchanged from R2Gen [1] for comparison purposes. In the training phase, we applied the following processes to a given image: (1) Resize a given image to 256 × 256. (2) Crop the image at a random location. (3) Flip horizontally the image randomly with a given probability, which is 0.5 in our experiment. Our model has a total number of 88.4 M parameters.

### 4.2. Datasets

We conducted our experiments on the following two datasets:IU X-ray [15]: a public radiography dataset collected by Indiana University with 7470 chest X-ray images and 3955 reports. With Institutional Review Board (IRB) approval (OHSRP# 5357) by the National Institutes of Health (NIH) Office of Human Research Protection Programs (OHSRP), this dataset was made publicly available by Indiana University. Most reports have front and lateral chest X-ray images. We followed Li et al. [9] to exclude the samples without reports. The image resolution was 512 × 512 and it was transformed to 224 × 224 before being used as the input.Chest X-ray14 [16]: Chest X-ray14 consists of 112,120 frontal-view X-ray images of 30,805 unique patients with fourteen common disease labels, mined from the text radiological reports. It is also provided with approval from the National Institutes of Health Institutional Review Board (IRB) for the National Institutes of Health (NIH) data.

Both datasets are partitioned into train/validation/test set by 7:1:2 of the entire dataset, respectively.

### 4.3. Pre-Training the GLVE

Pre-training refers to a model with a task, which is different from the downstream task. Several computer vision studies have shown that transfer learning from large pre-trained models is effective [17,18]. To better extract features of images, it is necessary to pre-train visual extractors that capture the features. Therefore, we used the Chest X-ray14 [16] dataset to pre-train GLVE. Given the Chest X-ray14 image, two different CNNs take in each $X_{global}$ and $X_{local}$ as input. The fully connected layer after the CNNs produces probability distributions of concatenated outputs of these CNNs belonging to each of 14 thorax diseases, and it is trained to minimize the binary cross-entropy loss, following Guan et al. [13]. We iterated the pre-training of the GLVE with 100 epochs and stopped training when the validation loss had stopped improving over 30 epochs. To demonstrate the effect of pre-training of the GLVE, we compared the performance when using pre-trained GLVE and when using not pre-trained GLVE, using the same architecture as R2Gen, except for the visual extractor. As shown in Table 1, we observed that the performance with pre-trained GLVE outperforms the not pre-trained GLVE, because not only the CNNs in the GLVE but also the mask that crops a distinct area of the global images are improved by gaining prior knowledge of the medical domain.

### 4.4. Evaluation

We performed our experiments with R2Gen (a baseline model in this study) [1], R2Gen + pre-trained GLVE, R2Gen + CEDT, and R2Gen + pre-trained GLVE + CEDT. R2Gen is a transformer-based model that learns, stores, and uses patterns of medical reports through memory. There has been a lot of transformer-based image captioning research which has shown good performance. Among them, R2Gen is a model that generates high quality reports in the medical field. We conducted the study by setting the R2Gen model as a baseline because it is a highly scalable model as it consists of variations of the standard transformer without any specialized methods which limit variation to the model, while the memory structure is suitable for the patterned description present in the X-ray medical report. We proposed methods based on R2Gen to improve the text generation for global features which is the weakness of transformer-based image captioning models. Therefore, we conducted our experiments to evaluate the performance of our methods, changing the method based on our method and the R2Gen model that was the baseline for our study. CoATT and CMAS-RL are other medical image captioning models that perform well. We refer the performance of these papers to compare our models. CoATT is a LSTM-based model and CMAS-RL is a cooperative multi-agent system.

The performance of all models used in the experiments was evaluated by three standard natural language generation (NLG) metrics. We utilized bilingual evaluation understudy (BLEU) [19], metric for evaluation of translation with explicit ordering (METEOR) [20], and recall-oriented understudy for gisting evaluation (ROUGE-L) [21]. BLEU is a metric to measure the performance of deep learning model for natural language processing by comparing the machine generation results with the exact results directly generated by humans under the main idea, “The closer a machine translation is to a professional human translation, the better it is”. BLEU-N means BLEU score using n-grams. Meteor is a metric which evaluates natural generation tasks by computing similarity based on the harmonic mean of unigram precision and recall by aligning machine generation to human generation. Rouge-L is a metric that evaluates natural language processing tasks comparing the longest matching part between machine generation and ground truth using the longest common subsequence (LCS). These conventional NLG metrics are used in text generation evaluation such as translation and summary. The performance of the R2Gen model is a reproduced result with the author’s code (https://github.com/cuhksz-nlp/R2Gen, 29 November 2021).

Table 2 summarizes the performance of all experimental results and of other models. As can be seen from Table 2, we conducted our experiments by applying our methods one by one and finally by applying all the methods. Overall, our models outperformed the baseline. Two experiments (R2Gen + GLVE, R2Gen + CEDT) which applied each of our methods to the baseline showed better performance than the baseline. This indicates that each of our methods is a meaningful method, showing performance improvement. R2Gen + GLVE showed good overall performance for metric, but the worst performance for BLEU-4. However, R2Gen + CEDT showed the best BLEU-4 score. CEDT showed strength in continuous text generation and made a good quality report when analyzing the generated results directly. This example is additionally analyzed below based on Figure 6. In addition, R2Gen + pre-trained GLVE + CEDT showed the best scores, by 5.62% in BLEU score, by 0.56% in MTR, and 1.98% in ROUGE-L compared to R2Gen. When the GLVE and the CEDT are combined, the overall score was the highest and the overall text was well generated. Our methods were well combined and showed higher performance than when only each method was applied alone. GLVE’s global and local features stored at various levels of the CEDT’s encoder were used without being lost during the decoding process. As a result, our full model showed great performance. However, as for the quality of the generated report, R2Gen + CEDT was the best. We found that pre-trained GLVE sometimes generates repetitive patterns that often appear in the training dataset. It was found that the diversity of the generated text was slightly reduced. This shows the weakness of GLVE in association, with BLEU-4 being the lowest.

Figure 6 shows four examples of chest X-ray images from the dataset. The baseline results are generated with R2Gen. Compared to the baseline, R2Gen + CEDT captures osseous structure or abnormal sizes from the global context information, which indicates that the proposed model makes use of both global and local features. For example, in the first, second, and fourth examples of Figure 6, the proposed model generates information about bones compared to the baseline model’s prediction, resulting in information closer to ground truth. In addition, in the first and fourth examples, the text is generated for abnormal sizes such as “upper limits of normal” and “not significantly enlarged”. These generating examples show the strength of our method for global features.

### 4.5. Ablation Study on the GLVE

We conducted several ablation studies on the GLVE to verify the contribution of each method. As shown in Table 3, we demonstrate the importance of the GLVE that extracts both global features and local features of a given image by reporting the performance while changing the input of the visual extractor. There are three different visual extractors for evaluating the importance of the GLVE: (1) Global-only visual extractor (GOVE), which takes only a given image as an input. (2) Local-only visual extractor (LOVE), which takes only cropped image with a binary mask to crop an important part of the image. (3) Pre-trained GLVE. In Table 3, R2Gen + Pre-trained GLVE shows better performance than R2Gen + GOVE and R2Gen + LOVE. Challenges in learning the features only from the global image arise from the noise included in the features and the various capture conditions, as mentioned in Section 3.1. On the other hand, despite being able to learn the features of the attention area (i.e., learned patches in Figure 2) among the global images through the local-only visual extractor, it may lead to loss of information, such as bone structure or organ size, that can be obtained from the global image. Therefore, pre-trained GLVE leads to better performance compared to GOVE and LOVE by learning rich representations of images because it can capture both global and local features, unlike other cases where only global or local features are learned.

## 5. Discussion

Automatic radiology report generation is an essential task in applying artificial intelligence to medical domains and it can reduce the burden on the radiologists. We proposed the GLVE and the CEDT to generate better reporting using both global and local features. Pre-trained GLVE not only learned which patches to pay attention to with the prior information of the medical domain, but it could capture both global and local information, unlike previous studies. In addition, the CEDT used all levels of information of the encoders to prevent the loss of this information during the decoding phase. Applying the above methodologies to a transformer-based medical image captioning model solved the problem of ignoring some information, such as organ size, bone structure, and lesion area, and improved performance in terms of the metric of natural language generation (NLG).

However, there are several problems in the methods as follows. First, it is apparent in the outputs of the model that certain expressions tend to be patterned. We observed that the patterned reports appear when pre-trained GLVE is included in the model architecture. One reason for the issue is that Chest X-ray14 [16] dataset has imbalanced data so that the GLVE learns biased information of specific organs, such as the heart and the lungs. On the other hand, RM of R2Gen [1], designed to remember information from the previous report generation process, is also the reason for this issue. Therefore, we would like to study ways to improve the quality of reports maintaining performance.

We thought that a specialized metric was needed to evaluate the performance of the medical image captioning, as we proceeded with the study. We first checked the performance through conventional NLG metric and performed the process of analyzing the generated report ourselves. However, in the process of analyzing the report, we found that there were parts where the score and our analysis results were different. It could not be said that the NLG metric score and the quality of the report were always directly proportional. This is because the organ observed in the radiology report on chest X-ray is limited, the method of describing diagnosis is patterned, and similar patterns exist in similar images, so there is a limit to measuring performance based on string matching of existing NLG metrics. In addition, there are several other issues about current medical image captioning metrics. For example, synonyms are not considered, such as describing the bone structure as “bone”, “skeleton”, and “osseous”. In addition, expressions such as “normal”, “unremarkable”, “grossly unremarkable”, “clear”, and “unchanged” show the same meaning but are not considered as the same expression. Medical reports are patterned into several expressions, including the above expression example. Therefore, these expressions can be summarized and considered as the same expression. In medical reports, subtle differences in size are important for the basis of medical judgment such as “enlarged”, “lower”, and “upper limits” while the presence or absence of disease name and symptoms is also important. The words for these meanings must have more contribution in evaluating the model compared to other words. Thus, to make further improvement and be used in practice, it is necessary to study a new radiology report specialized quantitative evaluation metric that considers the same expressions and more important words for medical reports with medical domain experts.

## 6. Conclusions and Future Work

In this paper, we proposed a novel report generation approach that can capture global features as well as local features by the GLVE and inject low-level and high-level encoders’ information by the CEDT. By using the GLVE, the proposed model captures both the features that can be obtained from a global image and from a local region. In addition, by using the CEDT, descriptions of features extracted from the GLVE are generated, leveraging all levels of information of the encoder. Our method outperformed the baseline model and generated better reports. For example, our model showed better performance by generating text for the global feature such as organ size and bone structure that the baseline could not produce. However, we found some reports contain repetitive patterns that exist in the training data. The generated reports tend to be more patterned as the diversity of text decreases. Although it shows high performance through the GLVE’s global and local features, it shows a weakness in that the quality of reports generated is diminished due to reduced diversity. We also found that NLG metrics have limitations in evaluating medical image captioning because of medical report’s characteristics. In future work, we plan to research improvements to prevent such an issue, providing a better description with more useful information and research a new medical report metric.

## Figures and Tables

**Figure 1 sensors-22-01429-f001:**
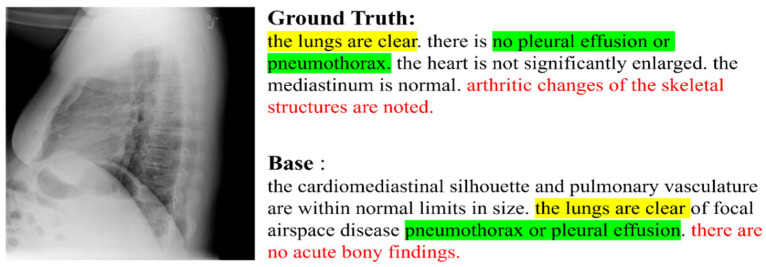
An example image and the caption generated from a baseline.

**Figure 2 sensors-22-01429-f002:**
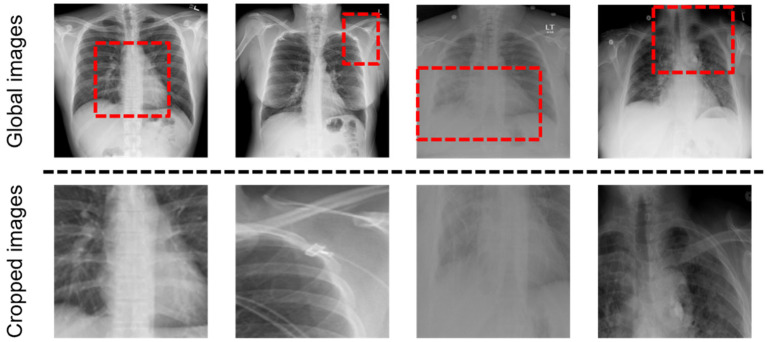
Example of input images of the GLVE. (**Top**): Global Radiology images, the input of the global visual extractor. We used a dashed box to indicate the region to be cropped. (**Bottom**): Cropped and resized images from corresponding images, used as the input of the local visual extractor.

**Figure 3 sensors-22-01429-f003:**
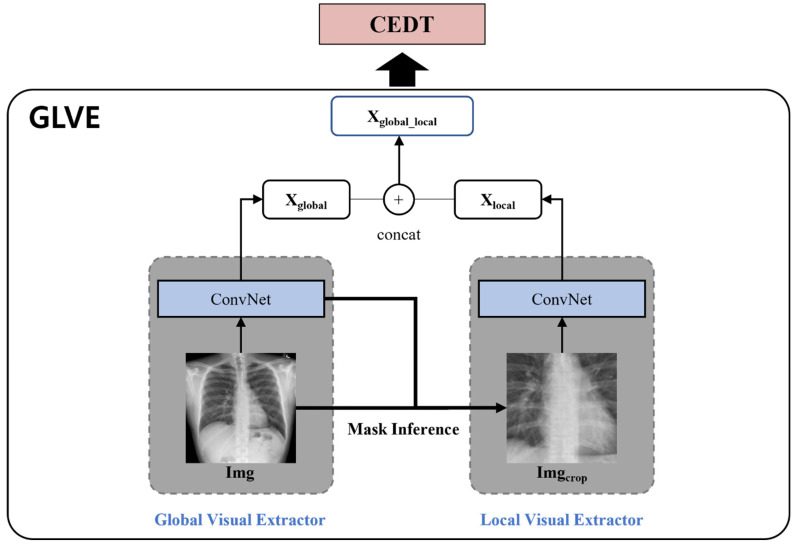
The pipeline of the GLVE. The GLVE consists of global visual extractor and local visual extractor. The global visual extractor uses a given image as an input and extracts global visual features. After extracting them, the attention guided mask inference process is performed to crop the image for the local visual extractor.

**Figure 4 sensors-22-01429-f004:**
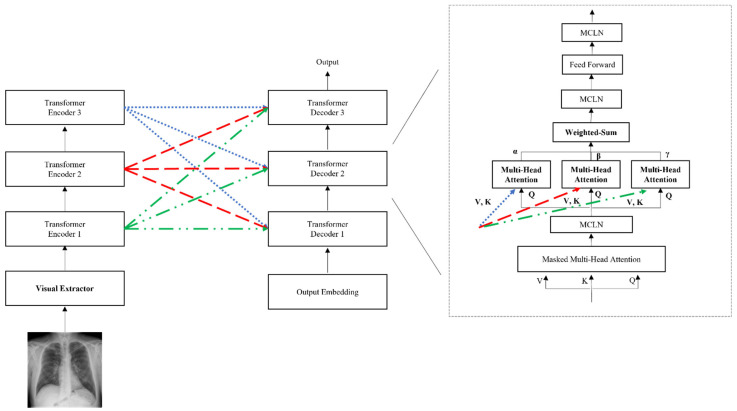
The proposed CEDT model architecture. The outputs of each encoder are used in the decoders.

**Figure 5 sensors-22-01429-f005:**
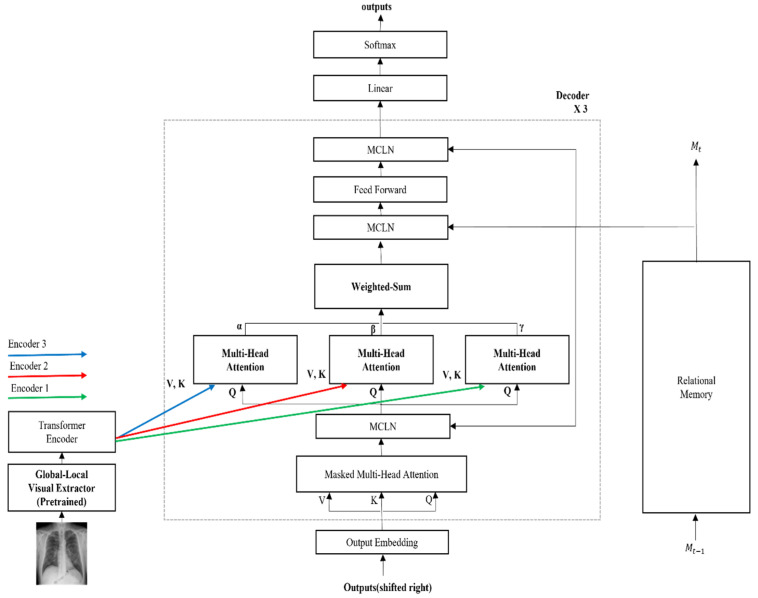
The overall architecture of our proposed model which combines both GLVE and CEDT. The gray dash box shows the detailed structure of our proposed decoder.

**Figure 6 sensors-22-01429-f006:**
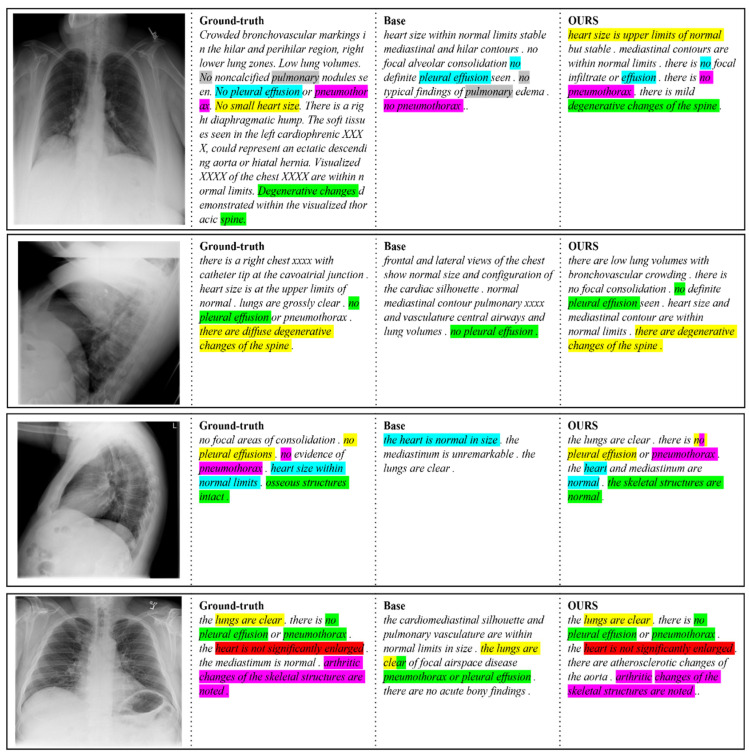
Ground-truth, the baseline report, and the report generated with the CEDT for two X-ray chest images. The same medical terms in the ground truth are highlighted with the same color in the generated text.

**Table 1 sensors-22-01429-t001:** The performance of the baseline with the GLVE and the baseline with pre-trained GLVE. One can observe that pre-trained GLVE leads to a better performance with the same condition.

R2Gen + GLVE				R2Gen + Pre-Trained GLVE			
BL-1	BL-2	BL-3	BL-4	BL-1	BL-2	BL-3	BL-4
0.4579	0.2941	0.2044	0.1476	0.4867	0.3002	0.2049	0.1484

**Table 2 sensors-22-01429-t002:** The performance of the baseline and proposed method against the IU X-ray dataset. BL-n denotes BLEU score using up to n-grams. MTR and RG-L denote METEOR and ROUGE-L, respectively. The experiment was conducted up to 5 times by changing the seed, and the table was filled out based on the highest score. Bold it to emphasize the best score for each metric.

MODEL	NLG METRICS					
	BL-1	BL-2	BL-3	BL-4	MTR	RG-L
R2Gen [1]	0.4502	0.2900	0.2124	0.1662	0.1868	0.3611
CoATT [11]	0.455	0.288	0.205	0.154	-	0.369
CMAS-RL [12]	0.464	0.301	0.210	0.154	-	0.362
Ours +Global-Local Visual Extractor	0.4867	0.3002	0.2049	0.1484	0.1888	0.3710
Ours +Cross Encoder-Decoder	0.4600	0.3020	0.2188	**0.1678**	0.1841	0.3544
Ours +Global-Local Visual Extractor +Cross Encoder-Decoder	**0.5064**	**0.3195**	**0.2201**	0.1603	**0.1924**	**0.3802**

**Table 3 sensors-22-01429-t003:** The baseline model with the GLVE outperforms both the model with the GOVE and the model with the LOVE in all BLEU scores.

R2Gen + GOVE				R2Gen + LOVE				R2Gen + Pre-Trained GLVE			
BL-1	BL-2	BL-3	BL-4	BL-1	BL-2	BL-3	BL-4	BL-1	BL-2	BL-3	BL-4
0.4566	0.2910	0.1967	0.1365	0.4266	0.2515	0.1767	0.1335	0.4867	0.3002	0.2049	0.1484

## Data Availability

Publicly available datasets were used in this study. The datasets can be found here: (1) IU X-ray (https://openi.nlm.nih.gov/detailedresult?img=CXR111_IM-0076-1001&query=&req=4, accessed on 28 November 2021) and (2) Chest X-ray14 (https://nihcc.app.box.com/v/ChestXray-NIHCC, accessed on 28 November 2021).

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
