# Peer review of "Cross Encoder-Decoder Transformer with Global-Local Visual Extractor for Medical Image Captioning"

_sensors, 2022, doi:10.3390/s22041429_

Round 1

Reviewer 1 Report

General comments:

Can the authors provide any background on relevant ethical considerations?

How do the presented results compare to human-level performance?

Please provide information regarding image resolutions. Both of the original images and those passed into the network.

Did the authors use any data augmentation techniques?

Specific comments:

(line 104) Is x_s^{global} a stand-in for every patch feature? If so, the subscript should not be "s", but probably something like "i for i from 1 to s".

(figure 2) It would be useful (though not entirely necessary) to show a box that indicates from where the lower images are cropped in the upper images.

(line 133) Should "as shown in Figure 2" actually reference figure 3?

(figure 3) Indicate the GLVE somehow. Maybe just a box around the entire pipeline?

(figure 4) It would be good to show how figures 3 and 4 relate to one another. Can they be unified? Also, it would be good to add references in those figure captions to the original papers for GLVE and CEDT, respectively.

(figure 4) Line colors are not visually-impaired or printer friendly. Perhaps use different line styles (and thicker lines).

(figure 5) Not referenced in text.

(figure 5) It is difficult to see how this figure contrasts with figures 3 and 4.

(lines 204-211) Formatting (spacing and indentation) for the bulleted list and surrounding paragraphs makes it difficult to read.

(section 4.2) Please provide training information. For examples, algorithm, epochs, stopping criteria, etc.

(section 4.3) Please provide context for why R2Gen was chosen as the baseline.

(line 222) You could provide a footnote to the "author's code."

(section 4.3) Provide a brief description of the metrics (BLEU, MTR, ROUGE-L) and their motivation.

(section 4.3) Please use consistency when referring to the models. For example, "R2Gen + pre-trained GLVE + CEDT" and "proposed model with the GLVE and the CEDT" refer to the same model.

(section 4.3) Are the reported scores for single experiments? If so, please train replicate models and report on means and standard deviations.

(table 1) Not referenced in text.

(analysis/discussion in section 4.3) I think this section must be expanded. Please provide additional information regarding the results. For example, some information and examples of the learned patches. Comparisons with other work must also be added.

Reviewer 2 Report

This work presents an automatic report generation approach, for medical image captioning, that can capture global features as well as local features; introducing a “Global-Local Visual Extractor” to capture both global features and local features and proposing a “Cross Encoder-Decoder Transformer” to generate captions by adding encoder’s low-level features and high-level features into the decoding process. The proposed method was evaluated on the IU X-ray dataset achieving better performance than the transformer-based baseline results.

The research was carefully designed, the conclusions are supported by the results, and the provided information is relevant for the knowledge field. Nevertheless, some issues should be addressed before this manuscript could be considered for publication (10 pages are not enough to present the complete information).

Full experimental details must be provided so that the results can be reproduced.

1) Detailed implementation information should be provided (hardware, software, configuration, settings).

2) Complete results should be presented and discussed.

The Conclusion section is superficial, should include advantages and disadvantages, limitation and recommendation for new implementations.

Reviewer 3 Report

The overall work is interesting. However, the quantitative analysis and scale invariant should be provided. Further improvement is listed below.

  1. Scale invariant for Global-Local Visual Extractor should be discussed;
  2. More multiple feature extraction and fusion should be compared;
  3. It would be good to highlight advantages and disadvantages in the abstract and main body of the paper.
  4. Further evaluation should be discussed.

Round 2

Reviewer 1 Report

The authors have done an excellent job addressing reviewer concerns. I still think the paper would benefit from an ethical discussion (not just as it pertains to the benchmark datasets), but maybe they disagree.

Reviewer 2 Report

The authors addressed the recommendations, the manuscript has been sufficiently improved and could be considered for publication.